# Token Identity as a Routing Signal for Residual MLP Experts

## Abstract

Learned mixture-of-experts routers use contextual hidden states to decide which parameters process each token. We ask how much of that routing signal can come from token identity alone when expert capacity is a small residual over a shared dense MLP. Our model maps each token ID to two narrow experts through a fixed, layer-specific lookup; the experts still transform the same contextual hidden state as the shared branch. In a matched single-seed comparison between 306.5M-parameter models trained on 8B FineWeb-Edu tokens, the token-routed model reaches loss 2.9329 versus 2.9482 for the dense baseline on the last common evaluation point. This stream is drawn from the training split, and the routed implementation is slower. The result is evidence that token identity can provide a useful residual routing signal in this run pair; learned-router comparisons, held-out evaluation, and replication across seeds remain open.

## 1 Introduction

Mixture-of-experts language models usually learn a router from each token's contextual hidden state (Shazeer et al., 2017; Fedus et al., 2022; Jiang et al., 2024). The router can adapt expert selection to context, but it also introduces parameters, balancing objectives, and dispatch decisions. Fixed routing provides a useful counterpoint: if a simple token-level signal allocates parameters effectively, some benefits attributed to contextual routing may instead come from stable conditional capacity.

We test this idea in a deliberately asymmetric design. Every token passes through a shared dense SwiGLU MLP. Token identity selects two additional experts, each much narrower than the shared branch, and their output is added as a gated residual. Selection is context independent, but expert computation is not: both the shared branch and the selected experts receive the contextual hidden state produced by the transformer. The experiment therefore asks whether token identity is useful for selecting a residual parameter subspace, not whether lexical lookup can replace contextual computation.

The primary comparison uses two 306.5M-parameter models trained for the same 8B-token budget. The models share their transformer backbone and stored MLP width; one uses a dense width of 4,096, while the other divides that width into a shared width of 3,840 and four routed experts of width 64, with two active per token. On the last common point of a fixed evaluation stream, the routed model reaches 2.9329 loss and the dense model 2.9482. The routed model trains more slowly in the current implementation, and the evaluation stream comes from the FineWeb-Edu training split.

This study contributes a controlled observation rather than a general verdict on expert routing. It defines a shared-plus-residual architecture in which token identity controls only expert selection, verifies the realized routing tables and capacity accounting, and reports the full matched-token trajectory. The single-seed result motivates direct comparisons with learned contextual routers and fixed random assignments; those comparisons are not part of the present evidence.

## 2 Related Work

Learned MoE routers use hidden states to select experts and commonly add an auxiliary objective or another balancing mechanism (Shazeer et al., 2017; Fedus et al., 2022; Jiang et al., 2024). These systems test the

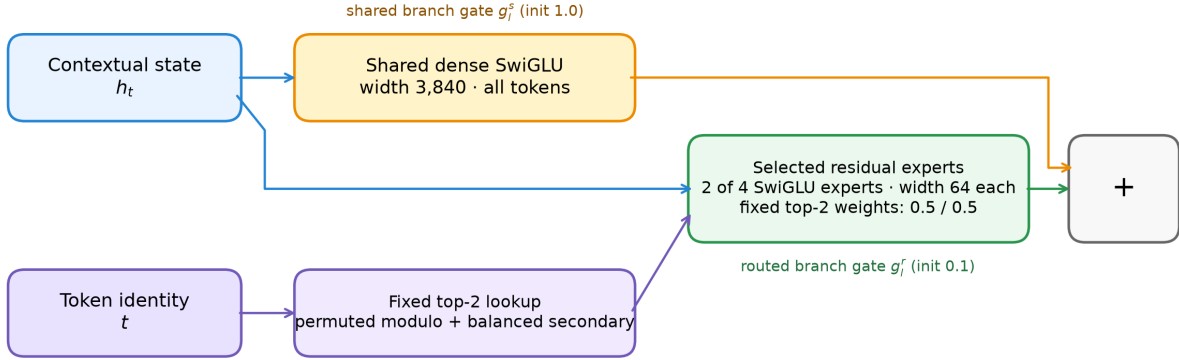

Figure 1: Token identity controls only the fixed top-2 lookup. The shared branch and selected residual experts receive the same contextual hidden state.

value of content-adaptive specialization. Our experiment tests a different axis: whether stable token identity is already informative when routing controls only a narrow residual branch.

Hash Layers is the closest precedent (Roller et al., 2021). It showed that fixed hashes of local token features can compete with learned routing in language models. Our design keeps a dense MLP active for every token and applies fixed top-2 lookup only to a small residual capacity. This makes the routed path a correction to common computation rather than the main feed-forward transformation.

Shared experts also appear in architectures such as DeepSeekMoE (Dai et al., 2024), where they capture computation common to all tokens. Statistical analyses of shared-expert mixtures provide results under explicit estimation assumptions (Nguyen et al., 2025). We use the shared-plus-routed decomposition as an architectural prior; our measurements do not test those theoretical guarantees.

## 3 COMPLEXITY-DEEP Architecture

### 3.1 Overview

The architecture changes only the feed-forward sublayer of a decoder-only transformer. Attention remains standard GQA with 16 query heads, four key/value heads, QK-Norm, RoPE, and causal scaled dot-product attention. Figure 1 isolates the changed computation. For token ID $t$ and contextual hidden state $\mathbf{x}$, the token ID selects residual parameters while both MLP branches transform $\mathbf{x}$.

### 3.2 Fixed token-identity routing

Each layer stores a fixed top-2 lookup. The primary expert is a seeded permutation $\pi_l$ of the token-ID residue:

$$r_{l,1}(t) = \pi_l(t \bmod E), \qquad E = 4. \tag{1}$$

The secondary table is constructed once, in ascending token-ID order, by selecting the least-used expert other than the primary:

$$r_{l,2}(t) = \arg\min_{e \neq r_{l,1}(t)} \sum_{u=0}^{t-1} \mathbb{1}[r_{l,2}(u) = e], \tag{2}$$

with lowest-index tie breaking. Because the 32,000-token vocabulary is divisible by four, each expert receives exactly 8,000 primary token IDs. This balances vocabulary cardinality, not corpus traffic.

### 3.3 Shared MLP and residual experts

For contextual hidden state $\mathbf{x}$ and token ID $t$, the feed-forward output is

$$\mathrm{MLP}_l(\mathbf{x}, t) = g_l^s \, \mathrm{Shared}_l(\mathbf{x}) + g_l^r \sum_{j=1}^{2} 0.5 \, \mathrm{Expert}_{r_{l,j}(t)}(\mathbf{x}), \tag{3}$$

where $g_l^s$ and $g_l^r$ are learned scalar gates initialized to 1.0 and 0.1. The selected expert outputs use fixed weights 0.5/0.5. Every branch is a SwiGLU transformation,

$$\mathrm{Expert}_i(\mathbf{x}) = (\mathrm{SiLU}(\mathbf{x}\mathbf{W}_{gate}^i) \odot \mathbf{x}\mathbf{W}_{up}^i)\mathbf{W}_{down}^i \tag{4}$$

and the shared branch has the same form with its own parameters. Token identity selects functions, but those functions operate on contextual states. The language-model objective is applied after the complete network; there is no independent expert objective.

### 3.4 Capacity and sparse dispatch

Let $d$ be the model dimension, $d_s$ the shared width, $d_e$ the width of each routed expert, $E$ the number of stored experts, and $k$ the number selected per token. Ignoring biases, the routed MLP stores

$$P_{\mathrm{TR}} = 3d(d_s + Ed_e) \tag{5}$$

parameters per layer and evaluates intermediate width

$$d_{\mathrm{active}} = d_s + kd_e \tag{6}$$

per token. In the evaluated model, $(d_s, E, d_e, k) = (3840, 4, 64, 2)$: stored width is 4096, matching the dense baseline, and active width is 3968. The implementation groups tokens by their fixed assignments and evaluates only the selected routed experts. Small expert kernels and dispatch still add runtime overhead, measured in Section 5.1.

## 4 Experimental Design

### 4.1 Primary comparison

We train a token-identity residual model and a dense baseline, both with 306.5M parameters. They use the same 18-layer backbone, tokenizer, 2,048-token context, FineWeb-Edu stream, optimizer, learning-rate schedule, BF16 precision, and 8B-token budget. The global batch is 1,048,576 tokens (eight GPUs, batch 64 per GPU), so equal training steps imply equal tokens seen. The controlled difference is the MLP parameterization in Table 1.

### 4.2 Optimization

Both models use AdamW with $(\beta_1, \beta_2) = (0.9, 0.95)$, weight decay 0.1, gradient clipping at 1.0, a 5% warmup, and cosine decay to 10% of the peak learning rate. Residual output projections use

$$\sigma_{\mathrm{residual}} = \frac{0.02}{\sqrt{2 \times L}} \tag{7}$$

for $L$ layers (Radford et al., 2019).

Table 1: Matched 300M model configurations.

| Parameter | Token-identity residual | Dense baseline |
|---|---|---|
| Layers ($L$) | 18 | 18 |
| Dimension ($d_{model}$) | 1024 | 1024 |
| Attention heads ($n_h$) | 16 | 16 |
| KV heads ($n_{kv}$) | 4 (GQA ratio 4:1) | 4 (GQA ratio 4:1) |
| Routed intermediate dimension | 256 total | — |
| Expert intermediate | 64 | — |
| Shared expert intermediate | 3840 | — |
| Experts ($n_{experts}$) | 4, top-$k = 2$ | 1 (dense) |
| Shared/routed gate initialization | 1.0 / 0.1 | — |
| Vocabulary | 32000 | 32000 |
| Max context | 2048 | 2048 |
| **Total parameters** | **306.5M** | **306.5M** |
| Active MLP path/token | shared + 2 routed experts | dense SwiGLU |

### 4.3 Routing and evaluation protocol

The saved tables in all 18 layers match Equations 1–2. No token-frequency table is present. We therefore identify the realized mechanism as permuted-modulo primary routing with a balanced secondary assignment. The fixed evaluation stream is drawn from the FineWeb-Edu training split; it is consistent across the two runs but is not an independent held-out set.

## 5 Results

### 5.1 Scaling Comparison (300M, 8B tokens)

The token-identity residual model begins behind the dense baseline and crosses it after roughly 0.8B tokens (Figure 2). At the last common evaluation point, step 7,500 or 7.864B tokens, its loss is **2.9329** versus **2.9482** for dense, a difference of $-0.0153$. At the final matched training point, step 7,620, the losses are **2.9231** and **2.9324**. Table 2 reports the trajectory at matched token counts.

Table 2: Training loss at matched steps and token counts. Negative differences favor the token-identity residual model.

| Step | Tokens seen | Dense (306.5M) | TR (306.5M, k=2) | Gap (TR − Dense) |
|---|---|---|---|---|
| 100 | 105M | 6.6698 | 6.8464 | +0.177 |
| 500 | 524M | 4.4450 | 4.4796 | +0.035 |
| 700 | 734M | 3.8567 | 3.8619 | +0.005 |
| 1000 | 1.05B | 3.5500 | **3.5324** | **−0.018** |
| 2000 | 2.10B | 3.1953 | **3.1720** | **−0.023** |
| 4000 | 4.19B | 3.0925 | **3.0738** | **−0.019** |
| 6000 | 6.29B | 3.0061 | **2.9906** | **−0.015** |
| 7620 | 7.99B | 2.9324 | **2.9231** | **−0.009** |

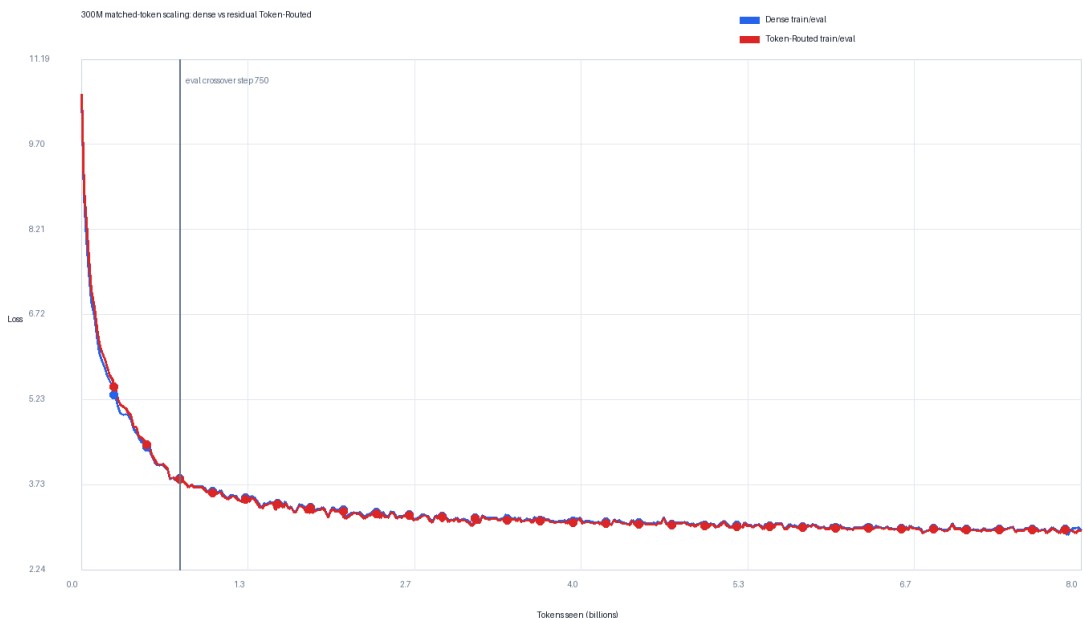

Figure 2: Training loss at matched token counts. The token-identity residual model (red) begins behind the dense baseline (blue), crosses after roughly 0.8B tokens, and remains ahead through the end of training.

**Routing traffic.** Final expert utilization is 0.2481/0.2635/0.2481/0.2403, with no dead expert. This is an observed property of the run, not a consequence of equal vocabulary cardinality.

**Training throughput.** On the same $8 \times$B300 system and global batch, dense training reaches approximately 0.95M tokens/s and token-identity residual training approximately 0.75M tokens/s. The reported quality comparison is matched by tokens, not by elapsed time.

### 5.2 100M Ablations

Seven single-seed models of approximately 100M parameters were each trained for 1.0003B tokens. The suite separates the shared branch, the residual branch, and several fixed top-2 assignments. The condition configured for frequency-aware routing contains no populated frequency table and therefore realizes modulo-primary/balanced-secondary lookup. Nominal modulo and round-robin controls have equivalent primary assignments in this setup, so differences between those rows are descriptive rather than causal.

Table 3: Matched-budget 100M ablations. Evaluation loss uses the fixed FineWeb-Edu training-split stream at step 750.

| Variant | Train @ 950 | Eval. @ 750 | Tok/s |
|---|---|---|---|
| Modulo-primary/balanced-secondary + shared | **4.8205** | **4.8887** | 321k |
| Dense residual | 4.8244 | 4.8958 | 397k |
| Modulo-adjacent top-2 + shared | 4.8320 | 4.9016 | 321k |
| Round-robin top-2 + shared | 4.8384 | 4.9072 | 321k |
| Random top-2 + shared | 4.8875 | 4.9577 | 318k |
| Shared-only | 4.9285 | 5.0012 | **402k** |
| Modulo-primary/balanced-secondary, no shared | 4.9693 | 5.0393 | 334k |

The shared branch is important in this suite: removing it gives the highest evaluation loss, while using it without routed residual experts also trails the dense and shared-plus-routed variants. The balanced-secondary shared run records the lowest loss, narrowly ahead of dense. Because each row is a single seed and some nominal controls share the same primary assignment, these results support the component comparison but not a causal ranking of fixed assignment strategies.

## 6 Discussion and Limitations

The result shows that a fixed token ID can select a useful residual parameter subspace in one matched run pair. It does not show that token IDs encode semantic expert classes. The assignments are modulo partitions up to a layer-specific permutation, and the selected experts transform contextual hidden states. A plausible interpretation is that stable token-conditioned capacity changes which functions are available to recurring lexical items while the shared MLP and attention preserve common contextual processing.

The experiment also does not isolate token identity from every alternative explanation. A matched non-routed residual, a fixed random partition, and learned contextual routers with standard balancing methods are needed to separate the value of identity-based assignment from residual capacity and conditional parameterization more generally. The 100M ablations in Section 5.2 include residual and random controls, but cannot provide a causal ranking of assignment strategies because some nominal controls realize equivalent primary assignments.

The 300M comparison uses one seed per architecture. Its evaluation stream comes from the training split, and standardized downstream or out-of-distribution evaluations are not available for the reported checkpoints. The routed implementation is slower than dense at matched batch. Finally, the evaluated lookup is not frequency aware: no frequency table is present in the checkpoints. These limits leave the central observation intact but constrain it to matched-token loss for this run pair.

## 7 Conclusion

Token identity provides a useful residual routing signal in the reported 306.5M-parameter run pair: after 8B training tokens, the shared-plus-routed model finishes ahead of a matched dense baseline. The result is small, single-seed, and slower in wall-clock training, but it poses a concrete question for learned MoE systems: how much contextual routing is necessary when a shared dense path already handles common computation?

### Broader Impact Statement

This work studies an architectural modification for language models. The submission does not include model weights; any future release should follow the applicable data, safety, and licensing requirements.

### Reproducibility Statement

The supplementary archive provides the PyTorch implementation, routing tests, run configurations, raw metrics, figure-generation scripts, and the 32k tokenizer used by the reported setup.

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
