# OpenReview forum: "Token Identity as a Routing Signal for Residual MLP Experts"
_TMLR — Under review for TMLR_

### Review · Reviewer_oB4k · 2026-07-06

**Summary Of Contributions:**

The paper proposes COMPLEXITY-DEEP, a language-model architecture centered on deterministic lexical routing for MLP blocks. Its strongest technical idea is a fixed token-id-to-expert routing table constructed by Zipf-balanced greedy bin-packing over empirical token frequencies. This directly addresses the difference between vocabulary-cardinality balance and expected corpus-load balance, and offers a simple alternative to learned routers and auxiliary load-balancing losses. The paper also introduces Mu-Guided Attention, where a per-layer state produced after the MLP is passed forward to modulate the next layer’s K, Q, V projections.

The ideas are interesting and potentially relevant to the TMLR audience. The current evidence falls short of the broader claims. The main issues are: the Mu-Guidance theorem analyzes a mechanism different from the architecture definition; the 187M ablation is confounded as a component study; and the 300M headline result evaluates a no-Mu, large-shared plus small-routed-residual variant, while the title and abstract emphasize the full architecture.

**Additional Comments:**

The current evidence supports a narrow claim: a single-seed, no-Mu, large-shared-MLP plus small gated lexical residual branch obtains a small matched-token train-loss advantage over dense while training at lower throughput. This is interesting preliminary evidence, though still short of the paper’s theoretical claims and broader thesis.

**Audience:**

No

**Audience Explanation:**

This paper is badly presented. The first 2 sections are just several lines, and some pages table and figure have too large blanks. I think readers will have a hard time in reading this paper.

**Broader Impact Concerns:**

Broader-impact issues appear limited to the usual risks of language models. The main barriers to acceptance are technical.

**Claims And Evidence:**

No

**Claims Explanation:**

1. The Mu-Guidance theorem analyzes a different mechanism from the submitted architecture. Eq. (11) However, Theorem 4.8 analyzes an EMA/pooling update, $\mu^{(l)}=\alpha \mu^{(l-1)}+(1-\alpha)\mathrm{Pool}(h^{(l)}),$ then states “in our implementation, (\alpha=0.9)” and gives a $0.9^{24}$ contraction factor. Eq. (11) contains a clamped learned parameter and a linear projection, whereas Theorem 4.8 uses an interpolation coefficient, a pooling operator, and a recurrence. This creates a mismatch with Eq. (11), the reported 18-layer 187M/300M models, and the 300M run whose Table 4 entry for Mu-Guidance is “No.” As a result, Theorem 4.8, Corollary 4.9, and the $E_\mu \leq \alpha^L |\mu^{(0)}|_2$ term in Proposition 4.12 leave the actual model unsupported. Even for the hypothetical EMA mechanism, the proof assumes away the dependence of $h^{(l)}$ on the previous Mu state in place of an explicit bound.

2. The 187M ablation is confounded as component evidence. Table 7 is labeled “Iso-parameter ablation,” but the dense baseline is 171M while the full Token-Routed+Shared+Mu model is 187M, about 9.4% larger. The same experiment is described inconsistently: Section 7.3/Table 8 say 500M tokens and 954 steps, while the Table 7 caption says “8B tokens FineWeb-Edu.” Figure 7 says “TR leads for 95% of training,” while the plotted gap crosses above zero late in training and stays in the red region where dense wins. Table 8 reports average loss, which is sensitive to early transients, and omits final train loss, final validation loss, and best validation loss. This ablation leaves unresolved whether Mu or the full model improves final quality over a matched dense baseline.

3. The 300M result evaluates a no-Mu residual-routed variant. The title and abstract emphasize Mu-Guided Attention, but Table 4 sets Mu-Guidance to “No” for the 300M Token-Routed model, and the limitations section lists full 300M Mu-guided scaling as future work. Moreover, Table 4’s Token-Routed model is dominated by a shared intermediate size 3840 branch, with only 256 total routed intermediate dimension, top-k = 2, and shared/routed gates 1.0/0.1. This experiment is best viewed as a large shared MLP plus a small gated lexical residual branch. It offers limited evidence for deterministic lexical routing as a replacement for dense MLPs or learned MoE routing. The final reported effect is small and single-seed: Table 9 gives a direct train-loss gap of -0.009 and a last-50 smoothed gap of -0.016. The same section reports lower throughput for Token-Routed training, about 0.75M tokens/s, than dense training, about 0.95M tokens/s, so the curves provide matched-token quality evidence, while compute- and wall-clock-efficiency evidence remains open.

**Requested Changes:**

1. Remove or rewrite Theorem 4.8, Corollary 4.9, and the (E_\mu) part of Proposition 4.12 to analyze the actual Eq. (11) Mu mechanism, or present them only as intuition.

2. Fix or rerun the 187M ablation as a true matched-parameter study; resolve the 171M/187M mismatch, the 500M-vs-8B inconsistency, and the Figure 7 caption conflict; report final train/validation losses.

3. Align title, abstract, and conclusions with the evaluated model: either provide a full 300M Mu-Guided run or frame the no-Mu 300M result as evidence for the residual-routed variant.

4. Add direct ablations: Zipf vs random/modulo/round-robin routing; shared-only and no-shared variants; lexical residual vs same-size non-routed residual; Mu vs matched QKV-bias/adapter/bypass controls.

5. Provide multi-seed 300M results with final validation loss and compute-normalized comparisons, including matched wall-clock and preferably matched FLOP curves.

---

> ### Author Response · Authors · 2026-07-07
> **Revision summary and response to Reviewer oB4k**
>
> Thank you for the detailed review. We substantially revised the submission to address the main mismatches you identified.
>
> First, we removed Mu-Guidance from the active contribution. The revised paper no longer frames Mu-Guided Attention as part of the evaluated architecture, title, abstract, or main claims. We also removed the Mu-focused theoretical claims from the active argument rather than presenting them as evidence for the submitted model.
>
> Second, we reframed the 300M result as what it actually evaluates: a no-Mu shared-plus-routed residual lexical variant. The revised paper explicitly states that most MLP capacity is in a shared dense branch, while a smaller Zipf-routed branch acts as a residual lexical specialization path. We no longer claim that deterministic lexical routing replaces dense MLP computation.
>
> Third, we added a controlled 100M, 1B-token ablation suite on $2\times$B200 with seven variants: dense residual, Zipf shared, Zipf no-shared, modulo shared, round-robin shared, random shared, and shared-only. This directly addresses the requested Zipf vs random/modulo/round-robin and shared-only/no-shared controls. In these runs, Zipf shared is the best variant and the only routed variant ahead of the dense residual baseline; shared-only and no-shared controls both underperform. The updated table reports final logged train loss, best validation loss, and throughput for every variant.
>
> Fourth, we clarified the evidence and limitations. The revised paper presents the result as matched-token quality evidence, not as a wall-clock training-efficiency claim. We report throughput separately and explicitly state that dense is faster in the current training implementation. We also acknowledge that the 300M result is single-seed and that multi-seed and compute-normalized comparisons remain outside the current evidence.
>
> Overall, the revised claim is intentionally narrower: frequency-aware deterministic lexical routing can be useful as a residual specialization path on top of a shared dense MLP backbone. We agree with the review that the previous version overemphasized full-architecture and Mu-guidance claims, and the revision removes that framing.

---

### Review · Reviewer_mBvo · 2026-07-16

**Summary Of Contributions:**

This paper introduces COMPLEXITY-DEEP, a mixture-of-expert variant, where the non-shared experts are chosen deterministically based on input token index and on current expert load. It is shown that for a given scale (300M) and training token budget (8B), COMPLEXITY-DEEP slightly outperforms other variants.

**Strengths**
- Fixating expert routing on token identity is an interesting experiment, and showing that performance can be on par with standard routing systems raises questions on the contextual/semantic nature of expert routing in MoE language models.

**Weaknesses**
- **The presentation of this paper is its most obvious flaw**. It looks to me that most -- if not all -- sections of the paper were written by a language model, as confirmed by a notorious AI detection system. It is not an issue in itself, but it leads to poor writing quality, and to several irrelevant comments. Providing **seven** significant numbers for the validation loss in the abstract is statistically meaningless and uninformative. Figure 2 is uninformative as it is a direct and trivial consequence of the method. Proposition 4.1 is not worthy of being written down. Most importantly, many unnatural comments seem to be left over from previous revisions, or the direct result of prompting a model to take reviews into consideration (e.g. "They do not establish a wall-clock efficiency advantage, an independent held-out validation result, or frequency-aware routing", " not as a theorem about the evaluated checkpoint", "The condition originally logged as “Zipf shared” is identified", ...).
- **Experiment scale & baselines:** the largest experiment scale consists in training 300M models on 8B tokens, and only compares the specific MoE variant with a dense baseline. This experiment scale is not very relevant in modern architectures. On B200 nodes, I hypothesize that these runs would cost <50 GPU-hours; in this context, running larger scale experiments should be possible. Moreover, some MoE baselines with different routers are missing.
- **Dubious/incomplete evaluation:** The smoothing over 50 final steps for the validation loss report is not justified. It would also be beneficial to report OOD perplexity or accuracy scores on easy benchmarks.

**Audience:**

No

**Audience Explanation:**

In the current state of the paper, the results are not insightful enough to provide a clear takeaway for the research community.

**Broader Impact Concerns:**

Not applicable.

**Claims And Evidence:**

Yes

**Claims Explanation:**

The claims are extremely factual and describe the technical results. The authors do not propose any broader conclusion. As such, the claims are accurate but they are also not very insightful.

**Requested Changes:**

- Improve presentation: The paper is most likely AI-written, and visibly results from corrections based on past reviews (some modifications are literally mentioned in the text). The statistical relevance of some results is not very convincing.
- The evaluation and experiments need to be more thorough. Report more compute-heavy experiments, add MoE baselines with standard routing approaches (at least load-balancing loss, and loss-free bias-balancing). Evaluate on more tasks.
- The inference sanity check should include a baseline.

---

> ### Author Response · Authors · 2026-07-17
> **Response to Reviewer mBvo: Presentation Revision**
>
> Thank you for the direct feedback. We agreed that the presentation obscured the main research question, so we substantially rewrote and reorganized the manuscript.
>
> The revised paper is now centered on a single question: can token identity usefully select a small residual parameter subspace when a shared dense path already handles common contextual computation? The title, abstract, introduction, architecture description, discussion, and conclusion have all been aligned with this framing.
>
> We also addressed the specific presentation issues raised in the review. The abstract now reports losses to four decimal places and no longer uses the final-50-step average as headline evidence. We removed the elementary balance proposition, its corresponding uninformative figure, repeated architecture descriptions, comments inherited from earlier revisions, and the inference benchmark that lacked a matched baseline. The new architecture figure makes clear that token identity controls only expert selection, while both the shared branch and the selected residual experts transform the same contextual hidden state.
>
> The seven matched-budget 100M ablations are now presented directly in the main Results section. The paper has also been reformatted as a six-page main submission, with references beginning on page 7.
>
> This revision addresses the writing, statistical reporting, and page-layout concerns in the review. It does not claim to address the requests for a larger-scale training campaign, learned-router baselines, multi-seed replication, or broader downstream and OOD evaluation. Those requests remain outside the evidence presented in the current revision.

---

> > ### Comment · Reviewer_mBvo · 2026-07-17
> >
> > This response and the underlying revision was submitted in less than 24 hours. It implies substantial changes to the presentation of the paper. I now suspect that not only this paper was entirely written by AI, but all subsequent revisions and responses were entirely delegated to AI based on reviewers' comments. It is especially concerning as it seems that some hallucinations have been introduced. Given the general low quality of the paper, I am opposed to such conduct and firmly recommend rejection.
> >
> > Were my assessment about AI usage incorrect, I still find the presentation unsatisfying:
> > - The NLL reported in the abstract is still the smoothed one;
> > - I can still find weird phrases such as "No token-frequency table is present", "The result is small"
> > - Information in the text of Section 5.1 and in Table 2 is inconsistent.
> > - My concerns about experimental design are unadressed. They also seem to have been the root cause of the initial rejection of this paper.
> >
> > I don't recommend uploading a fully AI-generated revision and response to this review, as I will stop interacting if I find it to be the case.

---

> > > ### Author Response · Authors · 2026-07-17
> > > **Clarification on AI-Assisted Writing and Author Responsibility**
> > >
> > > I understand the concern raised by the speed of the revision, and I want to clarify my process transparently. I initiated and directed this research project, designed the token-routed architecture, selected and ran the experiments, and made the scientific and experimental decisions. I used AI-assisted tools extensively to help turn my experimental records and instructions into English prose, to reorganize parts of the manuscript, and to assist with implementation and analysis. The project, the token-routing mechanism, and the experimental results were not autonomously invented or produced by the AI.
> > >
> > > However, the last revision was prepared too quickly, and I did not sufficiently verify every formulation and consistency before submitting it. That is my responsibility. I will not submit another rapid revision. I am auditing the manuscript against the raw experimental records and will remove or correct every statement that cannot be directly supported.

---

### Review · Reviewer_Paiy · 2026-07-20

**Summary Of Contributions:**

The paper proposes a feed-forward sublayer in which every token passes through a shared dense SwiGLU MLP while its token ID deterministically selects two of four narrow residual experts via a fixed, layer-specific lookup (a seeded permutation of the token-ID residue plus a balanced secondary table), with both branches transforming the same contextual hidden state and combined through learned scalar gates.

In a single-seed, width-matched comparison at 306.5M parameters over 8B FineWeb-Edu tokens, the token-routed model reaches 2.9329 loss against 2.9482 for a dense baseline on a fixed evaluation stream drawn from the training split, supported by a seven-condition 100M ablation suite, at a cost of roughly 21% training throughput.

**Strengths:**
- Claims are consistently qualified to "this run pair". The architecture is specified precisely enough to reimplement.
- The question is legitimate and well-positioned. Reducing routing to a context-free lookup over a small residual branch is a real narrowing of Hash Layers

**Weaknesses:**
- Figure 2 is not publishable as drawn. "COMPLEXITY-DEEP" appears exactly once, as a section heading, undefined and never reused.
- Train/eval terminology is inconsistent. Table 2's caption says "Training loss"; Section 4.3 and Table 3 describe a fixed evaluation stream from the training split; Figure 2's legend reads "train/eval." A reader cannot tell which quantity Table 2 reports.
- Table 3 compares train loss at step 950 against eval loss at step 750 with no stated reason for the mismatch.
- The motivation is under-delivered relative to its framing. The introduction proposes that "some benefits attributed to contextual routing may instead come from stable conditional capacity" and then never trains a learned router. The one comparison that would make the motivation bite is absent. What remains is "a fixed-lookup residual MoE marginally beats a width-matched dense MLP," which is narrower and less interesting than the setup promises.
- Experimental results are extremely shallow. Without ≥3 seeds, the primary result is not separable from noise, and the paper has no way to know either way.
- Evaluation is on the training split. It does not allow for any generalizations. And routing keyed to token identity is exactly the mechanism where memorization and generalization might diverge.
- No held-out perplexity and no downstream evaluations, so there is no way to know whether 0.009 nats corresponds to anything a user would notice.

**Audience:**

Yes

**Audience Explanation:**

MoE researchers may be interested in knowing the findings of this paper. However, this paper is badly presented.

**Claims And Evidence:**

No

**Claims Explanation:**

See Weaknesses

**Requested Changes:**

See Weaknesses